# Multi-Fused S,N-Heterocyclic Compounds for Targeting α-Synuclein Aggregates

**DOI:** 10.3390/cells14191531

**Published:** 2025-09-30

**Authors:** Chao Zheng, Jeffrey S. Stehouwer, Goverdhan Reddy Ummenthala, Yogeshkumar S. Munot, Neil Vasdev

**Affiliations:** 1MedChem Imaging, Inc., Boston, MA 02210, USA; 2Department of Radiology, University of Pittsburgh, Pittsburgh, PA 15213, USA; jeff.stehouwer@pitt.edu; 3Laxai Life Sciences, Hyderabad, Telangana 500078, India; goverdhanreddy.ummenthala@laxai.com (G.R.U.); yogeshkumar.m@laxai.com (Y.S.M.)

**Keywords:** α-Synuclein, PET, Parkinson’s disease, structure-activity relationships

## Abstract

The development of positron emission tomography (PET) tracers targeting α-synuclein (α-syn) aggregates is critical for the early diagnosis, differential classification, and therapeutic monitoring of synucleinopathies such as Parkinson’s disease (PD), dementia with Lewy bodies (DLB), and multiple system atrophy. Despite recent advances, challenges including the low abundance of α-syn aggregates (10–50× lower than amyloid-beta (Aβ) or Tau), structural heterogeneity (e.g., flat fibrils in PD vs. cylindrical forms in DLB), co-pathology with Aβ/Tau, and poor metabolic stability have hindered PET tracer development for this target. To optimize our previously reported pyridothiophene-based radiotracer, [^18^F]asyn-44, we present the synthesis and evaluation of novel S,N-heterocyclic scaffold derivatives for α-syn. A library of 49 compounds was synthesized, with 8 potent derivatives (LMD-**006**, LMD-**022**, LMD-**029**, LMD-**044**, LMD-**045**, LMD-**046**, LMD-**051**, and LMD-**052**) demonstrating equilibrium inhibition constants (*K*_i_) of 6–16 nM in PD brain homogenates, all of which are amenable for radiolabeling with fluorine-18. This work advances the molecular toolkit for synucleinopathies and provides a roadmap for overcoming barriers in PET tracer development, with lead compounds that can be considered for biomarker-guided clinical trials and targeted therapies.

## 1. Introduction

Synucleinopathies, including Parkinson’s disease (PD), dementia with Lewy bodies (DLB), and multiple system atrophy (MSA), are defined by the pathological accumulation of α-synuclein (α-syn) aggregates in neurons or glia [1,2,3]. These aggregates propagate in a prion-like manner, correlating with neurodegeneration and clinical progression [4]. Current diagnosis relies on nonspecific motor or cognitive symptoms, often delaying intervention until irreversible neuronal loss occurs [5]. The pathological progression of synucleinopathies is strongly associated with the accumulation of aggregated α-syn, spurring the development of α-syn-targeted positron emission tomography (PET) radiotracers as non-invasive biomarkers for early diagnosis, differential classification (e.g., distinguishing PD from MSA), and therapeutic monitoring. Such imaging agents would not only facilitate target engagement studies for α-syn-directed therapies but also enhance patient stratification in clinical trials [2,6,7,8,9,10] (Figure 1). The recent United States Food and Drug Administration (FDA) endorsement of α-syn seed amplification assays (αSyn-SAA) underscores the growing need for complementary in vivo imaging modalities to validate and extend biochemical diagnostic approaches, supporting a more comprehensive and precise management strategy for synucleinopathies [11].

Developing α-syn PET tracers faces formidable challenges: (1) α-syn aggregates occur at 10–50 times lower concentrations than amyloid-beta (Aβ) or tau, necessitating tracers with low- to sub-nanomolar affinity; (2) structural similarities between β-sheet-rich amyloid proteins demand high selectivity to avoid off-target binding; (3) post-translational modifications (e.g., phosphorylation at S129) and fibril polymorphism (e.g., flat vs. twisted conformations) complicate universal ligand design; and (4) intracellular α-syn inclusions require tracers to cross the blood–brain barrier (BBB) and cell membranes while resisting rapid metabolism [7,12,13,14,15,16,17]. Despite extensive efforts by academic laboratories and pharmaceutical companies, no α-syn tracer has achieved clinical validation for PD. PET radiopharmaceuticals (Figure 1) such as [^18^F]F0502B (binds α-syn fibrils with *K*_d_ = 10.97 nM) [18], [^18^F]ACI-12589 (selectively detects MSA pathology) [14], and [^18^F]C05-05 (binds DLB tissue with *IC*_50_ = 1.5 nM) have shown promise for imaging α-syn [19]. However, these compounds are limited by either insufficient selectivity or poor metabolic stability. More recently, the development of [^11^C]M503-1619, a tracer with high affinity for α-syn fibrils (*K*_i_ = 6.5 nM) and strong binding in PD brain tissue (*K*_d_ = 1.5 nM), has emerged as a promising radioligand for imaging α-syn pathology. Nonetheless, its rapid metabolic degradation may hinder its utility in long-duration scans and complicates quantitative imaging analyses using blood-input-based pharmacokinetic modelling [20].

Our recent efforts to develop PET radiotracers for α-syn have focused on diverse scaffolds such as phenothiazines, diphenylpyrazoles, and benzothiazoles. Among these, pyridothiophenes, exemplified by [^18^F]asyn-44, emerged as particularly promising due to their modular synthetic accessibility and the high binding affinity of [^3^H]asyn-44 (*K*_d_ = 1.85 nM) in post-mortem PD brain tissue, with minimal off-target interaction with amyloid-β (Aβ). Guided by cryo-EM structural insights into α-syn binding pockets, we previously synthesized a series of 47 derivatives to explore structure–activity relationships (SARs), evaluated their binding in PD brain homogenates, and performed preliminary in vivo PET imaging with [^18^F]asyn-44 [16]. These studies reinforced the potential of the pyridothiophene scaffold while also identifying radiometabolic instability in rodent brain as a potential limitation.

Herein, we report the synthesis and evaluation of a novel series of S,N-heterocyclic scaffold derivatives based on [^18^F]asyn-44. A focused library of 49 compounds, with the goal of being amenable for radiolabeling for PET, was synthesized, from which 8 derivatives showed high affinity for α-syn, with equilibrium inhibition constants (*K*_i_) ranging from 6 to 16 nM in PD brain homogenates. The most promising compound was evaluated for metabolic stability in human liver microsome assays as a potential substrate for efflux transporters to confirm pharmacokinetic properties suitable for in vivo PET imaging applications.

## 2. Materials and Methods

### 2.1. General

All ligands used in this investigation were obtained from MedChem Imaging, Inc. (Boston, MA, USA), each with a purity exceeding 95%. The radiolabelled compound [^3^H]Asyn-44 (specific activity: 1.5 GBq/μmol, concentration: 36 MBq/mL) was synthesized by Novandi Chemistry AB (Södertälje, Sweden) via tritiation of the corresponding dibromo precursor [16]. Additional reagents and chemicals were purchased from standard commercial sources and employed without further processing. Reaction progress was monitored through thin-layer chromatography (TLC) using silica gel 60 F25425 (Merck, Rahway, NJ, USA), with visualization achieved under 254 nm UV illumination. Compound purification was carried out using flash chromatography systems, including the Biotage Isolera One system (Uppsala, Sweden) and the Combi-flash Nextgen 100 (Teledyne ISCO, Lincoln, NE, USA). ^1^H-NMR spectra were obtained on a Bruker NEO-400 spectrometer (Bruker, Switzerland) with chemical shifts (δ, ppm) referenced to the deuterated solvent peak. Mass spectra data (*m*/*z*) were collected using a Waters Acquity SQD2 UPLC-MS system (Waters Corporation, Milford, MA, USA) operating in positive electrospray ionization (ESI+). Analytical and preparative HPLC analyses were conducted on a Waters 2695 Alliance (Waters Corporation, Milford, MA, USA).

### 2.2. Postmortem Tissues

Anterior cingulate cortex tissue from a Parkinson’s disease (PD) case was generously provided by Dr. Thomas Beach at the Banner Sun Health Research Institute in Arizona, USA. This sample showed abundant α-synuclein pathology without evidence of aggregated amyloid or TDP-43 inclusions. Preparation of the frozen blocks for binding assays followed previously established protocols [16,21].

### 2.3. In Vitro Competition (K_i_) Assays

The equilibrium inhibition constants (*K*_i_) for the unlabelled compounds were measured against [^3^H]asyn-44 using our previously reported procedure [16]. For these assays, frozen aliquots of homogenized human brain tissue (stored at −80 °C) were used. Homogenates were prepared at 10 mg/mL in phosphate-buffered saline, then diluted with 50 mM Tris buffer (pH 7.0) to a working concentration of 1 mg/mL.

Unlabeled competitor compounds were initially dissolved in DMSO at 400 μM, then diluted with Tris buffer to 20 μM, resulting in a 5% DMSO concentration. A serial dilution series (ranging from 6 μM to 4 nM) was prepared in the same 5% DMSO/Tris buffer solution to maintain consistent solvent conditions throughout the assay.

To perform the binding assay, 50 μL of the appropriately diluted competitor solution was mixed with 50 μL of [^3^H]asyn-44 (1 nM final concentration) and 800 μL of Tris buffer containing 20% ethanol and 0.1% bovine serum albumin (BSA). Then, 100 μL of the 1 mg/mL brain homogenate was added, bringing the final tissue concentration to 100 μg/mL. The reaction mixtures were incubated at room temperature for 60 min before filtration through Whatman GF/B glass fibre filters using a Brandel M-24R cell harvester (Gaithersburg, MD, USA). Filters were washed three times with 3 mL of Tris buffer containing 20% ethanol and 0.1% BSA.

After washing, retained radioactivity was measured using CytoScint-ES scintillation fluid following thorough vortexing. Specific binding was calculated by correcting for nonspecific binding (as described later), and the amount of bound radioligand was derived from the filter-associated radioactivity and the molar activity of the tritiated compound.

### 2.4. Two Point Screening Assays

Screening assays for the test compounds were performed at two concentrations—30 nM and 300 nM—using unlabelled competitors in human PD brain homogenates against [^3^H]asyn-44 as previously described [16]. To establish assay baselines, 0% inhibition was defined by the total binding observed in the absence of any unlabelled competitor, while 100% inhibition (representing nonspecific binding) was defined by the binding measured in the presence of 1 μM of the unlabeled radioligand. The % inhibition of the radioligand by the test compound at 30 or 300 nM was defined by the number of counts specifically bound at 30 or 300 nM divided by the difference in counts at 0% inhibition minus the counts at 100% inhibition multiplied by 100. Each assay was conducted in triplicate for both concentrations to ensure reproducibility and statistical reliability.

### 2.5. Metabolic Stability in Human Liver Microsomes

Metabolic stability assays were conducted using pooled human liver microsomes (HLMs) to evaluate the rate of in vitro metabolic degradation of the test compound. Incubations were carried out in duplicate (*n* = 2) at a final compound concentration of 1 μM. The protein concentration of the microsomal suspension was adjusted to 0.5 mg/mL using phosphate buffer (100 mM, pH 7.4). Reactions were initiated by the addition of NADPH (final concentration 1 mM) and incubated at 37 °C. Aliquots were taken at four time points: 0, 5, 15, and 30 min. Each reaction was quenched with ice-cold acetonitrile containing internal standard, followed by centrifugation to pellet proteins. Supernatants were analyzed using a validated LC-MS/MS method to determine the concentration of the parent compound at each time point. Verapamil was used as a reference compound to verify assay performance. Metabolic stability was assessed by calculating the percentage of the parent compound remaining at each time point. The elimination rate constant (k), half-life (t_1_/_2_), and intrinsic clearance (CLint) were derived from the natural log-transformed concentration vs. time data, assuming first-order kinetics.

### 2.6. In Vitro Transport Studies Using MDR1-Transfected MDCK Cells

MDR1-transfected MDCK cell monolayers, cultured for 5 days and obtained from the Netherlands Cancer Institute, were used for the in vitro bidirectional transport studies. Cells were maintained in Dulbecco’s Modified Eagle Medium (DMEM) supplemented with 10% fetal bovine serum (FBS) and 1% penicillin–streptomycin. They were subcultured every other day at a split ratio of 1:3 to 1:5. For transport experiments, cells were seeded onto polycarbonate Transwell^®^ filter inserts (Millipore) in 24-well plates at a density of 500,000 cells per well. After 24 h, the medium was replaced, and the cells were cultured for an additional 4 days to allow tight junction formation before initiating the transport assays. Transport studies were performed in duplicate using a 2 μM concentration of test compound, in both apical-to-basolateral (A→B) and basolateral-to-apical (B→A) directions. The apparent permeability coefficient (P_app) was calculated using the following equation: P_app_ (cm/s) = (Vr/C0) (1/S) (dC/dt)

Where

*P_app* is the apparent permeability,*V_r* is the volume of the receiver chamber,*C*_0_ is the initial concentration in the donor chamber,*S* is the surface area of the cell monolayer (0.7 cm^2^; for a 24-well insert),*dC*/*dt* is the rate of appearance of the compound in the receiver chamber.

Peak area ratio (PAR) was defined as the analyte peak area divided by the internal standard (IS) peak area.

The efflux ratio was calculated as: (1)$$Efflux\;ratio=\frac{Papp\;B-A}{Papp\;A-B}$$

Bioanalysis was carried out using LC-MS/MS. The system consisted of a Shimadzu Nexera X30 HPLC coupled with a Shimadzu LCMS-8045 Triple Quadrupole mass spectrometer (Shimadzu, Kyoto, Japan).

## 3. Results and Discussion

### 3.1. Chemical Synthesis

**Figure 2 cells-14-01531-f002:**
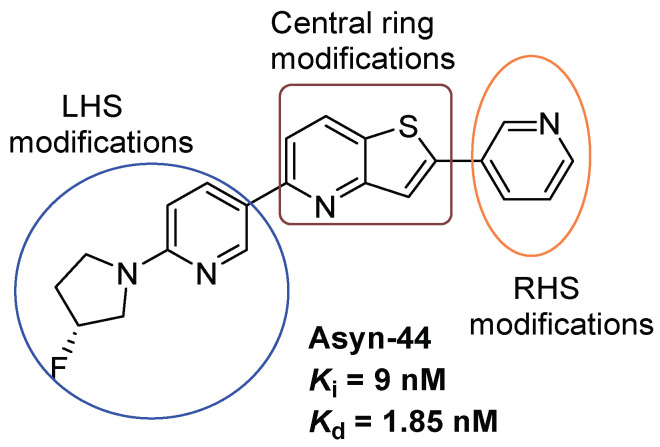
Design strategies for derivatization of Asyn-44.

Previously, we reported the discovery of a potent and selective pyridothiophene-based α-synuclein PET imaging radioligand, [^18^F]asyn-44. While [^18^F]asyn-44 [16] demonstrated favourable in vitro properties and acceptable brain uptake, the presence of brain-penetrant radiometabolites in rodents will require additional characterization in higher species prior to further progression in PET imaging studies. To advance our efforts in developing CNS-penetrant and selective α-synuclein ligands, herein we optimized the pyridothiophene scaffold guided by central nervous system multiparameter optimization (CNS MPO) scoring and key physicochemical parameters, including molecular weight, cLogD, number of aromatic rings, and topological polar surface area (tPSA). A focused library of S,N-heterocyclic derivatives of asyn-44, with the goal of being amenable for radiolabeling, was synthesized herein through systematic modifications at the central core, left-hand side (**LHS**), or right-hand side (**RHS**) of the scaffold (Figure 2), yielding three subclasses of compounds (as shown in the Appendix A).

Derivatives of subclass **I** were obtained from both **Int. IA** and **Int. IB**. **Int. IA** (Compound **3** in **Appendix A**) was synthesized in two steps from commercially available thiophen-3-amine and oxalic acid. **Int. IB** (Compound **4** in **Appendix A**) was prepared by bromination of **Int. IA**. Both **Int. IA** and **Int. IB** were subsequently coupled with boronic acids or boronic acid pinacol esters under palladium catalysis to form key intermediates. These intermediates were further transformed via Suzuki–Miyaura cross-coupling and/or Buchwald–Hartwig amination to afford a series of derivatives, including LMD-**001**, LMD-**002**, LMD-**009**, LMD-**011**, LMD-**049**, LMD-**045**, LMD-**005**, LMD-**013**, LMD-**060**, LMD-**006**, LMD-**014**, LMD-**062**, LMD-**015**, LMD-**024**, LMD-**026**, LMD-**036**, LMD-**039**, LMD-**040**, LMD-**041**, LMD-**063**, and LMD-**044**, with yields ranging from 0.5% to 86%.

To access compounds of subclass **II**, 5-bromo-1-tosyl-1H-thieno [3,2-c]pyrazole (**Int. II**, compound **61** in Appendix A) was synthesized from commercially available 2,5-dibromothiophene in three steps, following a patent (WO2023133229 A2 2023-07-13) reported protocol (Appendix A). Compound **6** was subsequently derivatized into a series of analogues. Most of these derivatives were obtained via palladium-catalyzed Suzuki–Miyaura cross-coupling reactions with thiazole and aryl boronic esters, affording intermediates that served as precursors for further functionalization. These intermediates underwent Buchwald–Hartwig amination with various aryl halide derivatives to yield the final products: LMD-**028**, LMD-**046**, LMD-**029**, LMD-**031**, LMD-**032**, LMD-**033**, LMD-**034**, LMD-**051**, LMD-**052**, LMD-**073**, LMD-**061**, LMD-**070**, LMD-**068**, and LMD-**069**, with yields ranging from 0.7% to 17%. In parallel, compound **60** was subjected to hydrodebromination and subsequent deprotection, followed by Buchwald–Hartwig amination to afford LMD-**031** with a yield of 1.5%. Alternatively, compound **60** underwent a palladium-catalyzed cross-coupling reaction followed by deprotection to yield 5-(1-methyl-1H-pyrazol-3-yl)-2H-thieno[3,2-c]pyrazole, which served as a key intermediate for a subsequent Buchwald–Hartwig amination. This process enabled the synthesis of a series of analogues, including LMD-**054**, LMD-**066**, LMD-**067**, LMD-**072**, LMD-**074**, LMD-**075**, LMD-**076**, and LMD-**077**, with yields ranging from 3% to 14%.

For the synthesis of subclass **III** compounds (Figure 3), the pyrrolo[3,4-b]pyridin-5-one core (**Int. III**) was constructed from commercially available 6-chloro-2-methylnicotinic acid via three-step reactions [22]. This included condensation with various amine derivatives, such as 1-methyl-1H-pyrazol-3-amine, 1-methyl-1H-imidazol-4-amine, and (4-methoxyphenyl)methanamine, to afford the corresponding substituted pyrrolo[4,4-b]pyridin-5-one intermediates. These intermediates were subsequently cross-coupled with commercially available aryl boronic esters via a Suzuki–Miyaura reaction to afford the final compounds LMD-**016**, LMD-**017**, and LMD-**023**, as well as key intermediates used in the synthesis of LMD-**064**, LMD-**019**, and LMD-**022**. The non-optimized yields for these new inhibitors ranged from 1% to 60%.

### 3.2. Binding Assays and SAR Studies

All screened compounds were >95% pure as determined by ^1^H-NMR and analytical HPLC (Appendix A). The in vitro evaluation of the synthesized derivatives was carried out to identify the most promising candidates for α-synuclein PET radiotracer development. The compounds underwent initial screening using a two-point assay, consistent with protocols described in prior work [16,21]. This preliminary screen aimed to identify structural variants capable of displacing [^3^H]asyn-44 in a dose-dependent manner, thereby prioritizing compounds for more detailed competition binding assays. Testing was performed in human PD brain tissue homogenates at two concentrations: 30 nM and 300 nM (Appendix A). While numerous compounds demonstrated strong displacement at the higher concentration (with <50% of [^3^H]asyn-44 remaining bound), only a limited subset exhibited comparable activity at 30 nM. For compounds exhibiting pronounced inhibition at 300 nM (≤15% bound [^3^H]asyn-44) and moderate to good inhibition at 30 nM (around 70% bound), full competition binding assays were conducted to determine their K_i_ values (Appendix A).

Among these, the SAR results are summarized in Appendix A Appendix A. The pyrazine analogue, LMD-**028**, showed improved potency compared to the pyridine analogue in LMD-**029**. The **N1** analogue in LMD-**046** demonstrated greater potency than the **N2** analogue in LMD-**028**. The piperidine ring in LMD-**033** showed enhanced potency relative to the pyrrolidine analogue in LMD-**051**. However, replacing the pyridine ring in Asyn-44 with a pyrazine ring in LMD-**033** resulted in reduced potency. Similarly, the **N2** analogue in LMD-**052** displayed decreased potency compared to the **N1** analogue in LMD-**073**. Substitution of the pyrrolidine ring in LMD-**052** with a piperidine ring in LMD-**068** also led to a loss in potency. In a comparison between Asyn-44 and LMD-**041**, shifting the pyridine ring from the **C-2** to the **C-3** position on the central scaffold in LMD-**041** resulted in diminished potency. Additionally, the pyridine ring in LMD-**041** was less effective than the N-methylpyrazole ring in LMD-**063**. LMD-**045** demonstrated improved potency due to the incorporation of a piperazine ring at the **LHS**, compared to LMD-**041**. Although both LMD-**026** and LMD-**027** showed reduced potency, LMD-**044** exhibited improved activity, likely due to the presence of a piperazine moiety at the **LHS**.

Asyn-44 and the thiazole analogue LMD-**006** were found to be equipotent. In contrast, the N-methylpyrazole analogue LMD-**005** showed decreased potency compared to Asyn-44. The 3-methylpyrrolidine group in LMD-**001** also led to reduced activity relative to the pyrrolidine analogue in LMD-**002**. Asyn-44 and LMD-**009** displayed comparable potency, despite the presence of a pyrazine ring at the **LHS** in LMD-**009**, which did not enhance activity. The replacement of the pyridine ring at the **RHS** of LMD-**009** with thiazole or methoxy-pyridine resulted in equipotency, while substitution with N-methylpyrazole (LMD-**013**) led to a loss in potency.

The piperidine analogue LMD-**060** showed improved potency compared to the pyrrolidine analogue LMD-**013**. Substitution with a pyrazine ring at the LHS in LMD-**022** and LMD-**017** led to better potency than the pyridine-containing analogues LMD-**064** and LMD-**016**. Modification of the central core in LMD-**019**, relative to Asyn-44, did not result in any improvement in potency. A significant drop in potency was observed for LMD-**064** in comparison with LMD-**016**, LMD-**019**, and LMD-**023**.

### 3.3. Metabolic Stability, Permeability, and Efflux Characteristics of Selected LMD Compounds

To assess metabolic stability, selected LMD compounds were incubated with human liver microsomes (h-MR), and their half-lives (t_1_/_2_) were determined (Table 1 and Figure 4). The results demonstrated a broad range of metabolic stability among the compounds. LMD-**022** (*K*_i_ = 16 nM) showed excellent stability with a half-life of 75 min. In contrast, compounds such as LMD-**006** (*K*_i_ = 12 nM, t_1_/_2_ = 6.81 min), LMD-**032** (*K*_i_ = 50 nM, t_1_/_2_ = 7.05 min), LMD-**044** (*K*_i_ = 9.1 nM, t_1_/_2_ = 6.37 min), and LMD-**051** (*K*_i_ = 6 nM, t_1_/_2_ = 7.6 min) showed rapid metabolic turnover, indicating limited stability. LMD-**066** (*K*_i_ = 11.8 nM) was found to be metabolically unstable under the assay conditions. Moderate stability was observed for LMD-**045** (*K*_i_ = 13.3 nM, t_1_/_2_ = 10.5 min) and LMD-**051** (*K*_i_ = 9.2 nM, t_1_/_2_ = 17 min), suggesting some potential for further optimization. (Chromatograms in the Appendix A.)

**Figure 4 cells-14-01531-f004:**
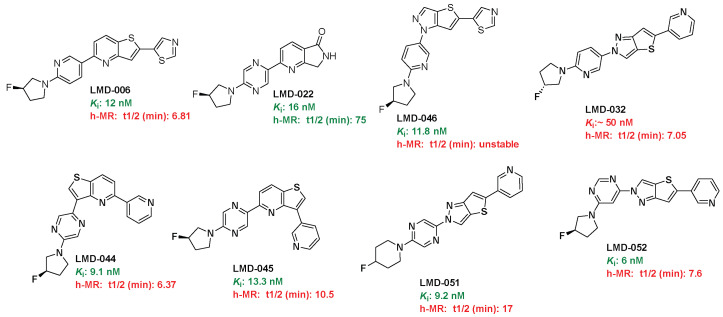
Active compounds.

Based on its favourable metabolic stability and binding affinity, LMD-**022** was selected for further evaluation of its permeability and efflux characteristics using a bidirectional Caco-2 cell assay (Table 2). LMD-**022** demonstrated high permeability, with an apical-to-basolateral (A→B) permeability of 40.6 × 10^−6^ cm/s and a basolateral-to-apical (B→A) permeability of 37.6 × 10^−6^ cm/s. The calculated efflux ratio was 0.93, indicating that LMD-**022** is not a substrate for active efflux transporters. For comparison, digoxin, a known P-glycoprotein substrate, showed low permeability (A→B: 1.65 × 10^−6^ cm/s) and a high efflux ratio of 14.5, confirming its classification as a substrate of efflux. In contrast, propranolol, a compound with moderate permeability (A→B: 9.28 × 10^−6^ cm/s) and an efflux ratio of 0.9, was similarly categorized as not being a substrate. These results suggest that LMD-022 possesses favourable permeability characteristics and is unlikely to be limited by efflux mechanisms in vivo. (Chromatograms in the Appendix A.)

**Table 1 cells-14-01531-t001:** The metabolic stability of selected LMD compounds in human liver microsomes.

Compound ID	% Remaining at 5 min	% Remaining at 30 min	t_1/2_ (min)	Clint (µg/min/mg)	Clearance Category ^2^
LMD-**032**	53.3	5.02	7.05	197	High
LMD-**046**	3.43	NC ^1^	NC	NC	NC
LMD-**051**	75.8	30.3	17.0	81.5	High
LMD-**052**	39.7	5.61	7.60	182	High
LMD-**044**	45.7	3.71	6.37	218	High
LMD-**045**	65.3	13.5	10.5	132	High
LMD-**006**	39.6	4.32	6.81	203	High
LMD-**022**	92.4	75.4	75.3	18.4	Moderate
Verapamil	43.6	4.64	6.89	201	High

^1^ NC: Not Computable. ^2^ As per Cyprotex Website.

**Table 2 cells-14-01531-t002:** Summary of MDR1-MDCK permeability and efflux data for LMD-022 and control compounds.

Compound ID	A→B (×10^−6^ cm/s)	B→A (×10^−6^ cm/s)	Efflux Ratio (B→A/A→B)	Permeability Category	Efflux Category
LMD-**022**	40.6	37.6	0.93	High	Not a substrate
Digoxin	1.65	24	14.5	Low	Substrate of efflux
Propranolol	9.28	8.03	0.9	Medium	Not a substrate

**Table 3 cells-14-01531-t003:** Inhibition constant (*K*_i_) values of LMD-022 in post-mortem Alzheimer’s disease (AD) brain tissue.

Radioligand	*K*_i_ (nM)
[^3^H]PiB	>10,000
[^3^H]MK-6240	>10,000
[^3^H]Z-2340	>10,000
[^3^H]PI-2620	2764

We assessed the binding affinity of LMD-022 using a radioligand competition binding assay in post-mortem PD brain tissues. LMD-022 competed effectively with [^3^H]Asyn-44, yielding a *K*_i_ value of 16 nM, indicating high binding affinity to its target in PD tissue (Table 3). To evaluate potential off-target binding in AD brain tissues, we conducted additional binding assays using a panel of AD-related radioligands, including [^3^H]PiB, [^3^H]MK-6240, [^3^H]Z-2340, and [^3^H]PI-2620. The lack of appreciable binding of LMD-022 with the tritium-labelled radioligandss observed in AD tissues across these assays, suggests high selectivity for PD pathology over AD-related targets. Our future work will aim to optimize chemical yields of lead compounds, including **LMD-022**, and carry out α-syn fibril seed amplification assays or immunohistochemical co-localization with pathologic α-syn to verify target engagement as well as radiolabel lead molecules for in vivo PET imaging studies.

## 4. Conclusions

In this study, we designed and synthesized a diverse library of 49 S,N-heterocyclic derivatives based on the pyridothiophene scaffold of the previously reported α-synuclein ligand [^18^F]asyn-44. Through systematic structural modifications across three subclasses—thieno[3,2-b]pyridine, 6,7-dihydro-5H-pyrrolo[3,4-b]pyridin-5-one, and 2H-thieno[3,2-c]pyrazole—we identified eight compounds (LMD-006, LMD-022, LMD-029, LMD-044, LMD-045, LMD-046, LMD-051, and LMD-052) with high affinity (*K*_i_ = 6–16 nM) in PD brain tissue homogenates. Among these, LMD-022 emerged as a lead candidate, exhibiting potent binding (K_i_ = 16 nM), excellent metabolic stability (human liver microsome t_½_ = 75 min), and favourable pharmacokinetic characteristics. Its high permeability (Caco-2 P_app_ = 40.6 × 10^−6^ cm/s) and lack of efflux liability (efflux ratio = 0.93) further support its potential as a brain-penetrating radioligand. Ongoing studies are focused on radiolabeling LMD-022 and related analogues and evaluating their in vivo performance through preclinical PET imaging.

## Figures and Tables

**Figure 1 cells-14-01531-f001:**
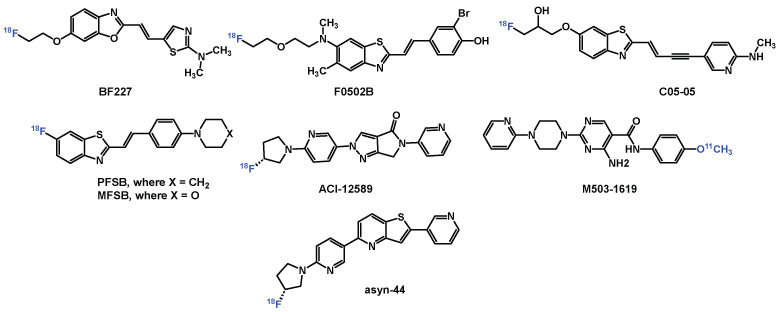
Selected PET tracers for imaging α-syn.

**Figure 3 cells-14-01531-f003:**
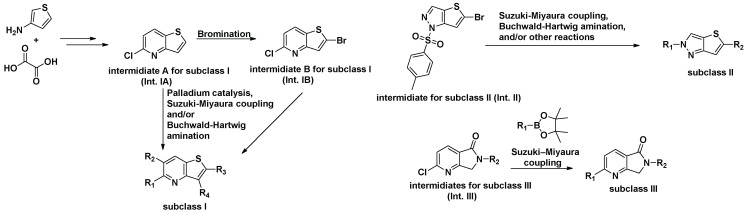
Three Subclasses of Novel Derivatives and their synthetic strategies.

## Data Availability

The original contributions presented in this study are included in the article/Appendix A. Further inquiries can be directed to the corresponding author.

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
