# Peer review of "Multi-Fused S,N-Heterocyclic Compounds for Targeting α-Synuclein Aggregates"

_cells, 2025, doi:10.3390/cells14191531_

Round 1
Reviewer 1 Report
Comments and Suggestions for Authors
The article by Chao Zheng et al. presents a highly comprehensive SAR study on the lead compound [¹⁸F]asyn-44. The scope of the work and the number of derivatives explored are impressive. Additionally, the methodologies and experimental procedures are clearly and thoroughly described. Overall, I would recommend this manuscript for publication in Cells after minor revision. Below are a few specific suggestions for improvement:
- Figure 1: Please ensure consistency in the depiction of methyl groups. In some structures, they are shown as "-CH₃", while in others they are not explicitly drawn. A uniform style would improve clarity.
- Compound numbers throughout the manuscript should be consistently formatted in bold to enhance readability.
- I recommend moving the synthetic schemes into the main article. As currently presented, it is difficult to follow the synthetic strategies. A general scheme for each class of compounds would suffice and greatly help the reader.
- Some reported yields are extremely low. While I understand these syntheses were not optimized, a brief explanation would provide helpful context.
- Please consider adding a figure summarizing the SAR findings. A visual overview would significantly enhance the reader's understanding of the structure–activity trends.
- If feasible, please include 13C NMR data for at least the final compounds in the Supporting Information.
Reviewer 2 Report
Comments and Suggestions for Authors
The study addresses a major unmet need to develop PET radioligands for α-synuclein aggregates, crucial for early diagnosis and monitoring of Parkinson’s disease (PD) and related synucleinopathies. It builds on [18F]asyn-44, a previously reported scaffold, and aims to overcome known issues with metabolic instability and selectivity. In the study a focused library of 49 S,N-heterocyclic derivatives was synthesized via three scaffold diversification strategies, with logical modifications to optimize binding, CNS permeability, and radiolabeling potential. Systematic SAR evaluation is well conducted, with structure–activity insights clearly interpreted.
Although it is interesting, there are some major drawbacks need to be addressed by the authors:
1, The study lacks in vivo PET imaging or biodistribution results, even for the most promising compound (LMD-022). Without animal data, conclusions about radiotracer suitability are premature. The authors are suggested to include or plan rodent PET studies to assess brain uptake, distribution, and target specificity.
2, The selectivity of compounds against amyloid-β (Aβ) and tau aggregates is only briefly mentioned and not experimentally demonstrated in this study. The authors are suggested to conduct cross-competition binding assays in Aβ- and tau-rich brain homogenates to evaluate off-target binding.
3, In the study, many key compounds (e.g., LMD-006, LMD-045, LMD-046) are reported with very low synthetic yields (0.5%–17%), which may challenge scalability for radiochemistry and clinical translation. The authors should provide improved synthetic routes or comment on future optimization strategies.
4, Although binding affinity is shown in PD brain homogenates, the biological specificity to pathogenic α-syn species (vs. physiological α-syn or non-fibrillar forms) is not addressed. The authors should use α-syn fibril seed amplification assays (SAA) or immunohistochemical co-localization with pathologic α-syn to verify target engagement.
Minor defects:
1, Tables summarizing Ki values, metabolic half-lives, and SAR findings are mostly relegated to Supplementary Information, limiting quick access for readers. The authors are suggested to move key tables (e.g., for top 8 compounds) and SAR summary into the main text for better impact.
Reviewer 3 Report
Comments and Suggestions for Authors
Dear authors,
While the manuscript might be of interest for many readers, some aspects are missing or need to be addressed mandatory (major revision required):
- The iThenticate report shows 24% similarity (too high)! Please rephrase that parts.
- There is no discussion section. You should comment on the results of your work and compare them to the existing literature.
- You have to mention the limitations of your study.
- There is no mention of ethical consent in the text. please include it in a particular subchapter in the materials and method section.
- The abstract should be structured.
- The conclusion section should include an additional paragraph about the future research directions.
Round 2
Reviewer 2 Report
Comments and Suggestions for Authors
The authors have revised their manuscript and provide new evidence, which can improve the quality of the study. However, The authors refuse to perform experiments to check the biological specificity to pathogenic α-syn species. As this work is focus on PET approaches targeting synuclein aggregates. How the tracers interact with aggregated or non-aggregated synuclein is the major evidence for their study. Therefore new evidence on syn aggregates must be presented.
Author Response
Comments 1: The authors have revised their manuscript and provide new evidence, which can improve the quality of the study. However, The authors refuse to perform experiments to check the biological specificity to pathogenic α-syn species. As this work is focus on PET approaches targeting synuclein aggregates. How the tracers interact with aggregated or non-aggregated synuclein is the major evidence for their study. Therefore new evidence on syn aggregates must be presented.
Reviewer 1: We thank the Reviewer for raising this point, and would like to mention that we did not refuse to do this work and these experiments are planned in the future. The proposed studies require [3H]LMD-022, which will be synthesized, and evaluated in the future, but is not yet available. The studies would be carried out similarly to our [3H]asyn-44 autoradiography and immunohistochemical studies in Chem. Eur. J. 2024,30,e202303921.
Reviewer 3 Report
Comments and Suggestions for Authors
To whom it may concern,
Thank you for resubmitting the revised version of your work. I appreciate your responses to all my inquiries. The final version of this responds to the quality and originality criteria of this journal. I have no further comments.
Author Response
Comments 1:
Thank you for resubmitting the revised version of your work. I appreciate your responses to all my inquiries. The final version of this responds to the quality and originality criteria of this journal. I have no further comments.
Response 1: We thank the Reviewer for their support of our work and publication in Cells.
Round 3
Reviewer 2 Report
Comments and Suggestions for Authors
The authors have addressed most of concerns and provide new evidence. However, they refuse to investigate whether their compounds can have any influence on SAAs. The SAAs is the vital assay to verify the inhibition of synuclein aggregation by anti-PD therapeutic compounds. I do not agree that the assay should be performed in the future study. I insist that the authors should provide evidence whether their synuclein protein aggregation inhibiting compounds can be positive in SAAs study. Without these evidence, their potential therapeutic significance for PD can not be assessed.